# Is REM Density a Measure of Arousal during Sleep?

**DOI:** 10.3390/brainsci13030378

**Published:** 2023-02-21

**Authors:** Giuseppe Barbato

**Affiliations:** Department of Psychology, Università degli Studi della Campania, Luigi Vanvitelli, Viale Ellittico 31, 81100 Caserta, Italy; giuseppe.barbato@unicampania.it

**Keywords:** REM density, sleep depth, arousal, depression, phasic REM

## Abstract

Rapid eye movements (REMs), an expression of REM sleep phasic activity, occur against a stable background of cortical desynchronization and the absence of axial tone. The significance of REMs during the sleep period was initially attributed to the mental content of dreams, linking the REMs to the dream scenario. Although fascinating, the so-called “scanning hypothesis” has not been supported by consistent evidence, and thus an alternative hypothesis is necessary to understand REMs significance during sleep. Some data suggest that the frequency of REMs during the REM sleep period, known as REM density, might be related to sleep depth or arousal during sleep. REM density increases across the night concomitantly with the progressive reduction in sleep pressure, and consistently it is higher at the circadian time when arousal appears to be higher, and it is decreased in those conditions, such as after sleep deprivation, which produce increased sleep pressure. REM density is also increased in major affective disorders, and it has been suggested either as a risk factor to develop the illness or as a predictive index of response to drug treatment. Disfunction of the neurotransmitter systems involved in arousal mechanisms and wake/sleep control might underlie the altered REM density described in depression. Understanding of the REM density mechanisms could help to untangle functional significance and regulation of REM sleep. Following the seminal idea of Aserinsky that REM density is an index of sleep satiety, it may also provide a sensitive measure of sleep homeostasis in addition to, or even as an alternative to, the consolidated analysis of slow wave activity. REM density can also be utilized to explore those mechanisms which end sleep, and considered a physiological marker which indicate during sleep the “time to wake”.

## 1. Introduction

Discovery of rapid eye movements (REMs) during sleep was the imprimatur of a “Copernican revolution” in our understanding of sleep and of concomitant mental activity. In their seminal work, Aserinsky and Kleitman [1] reported that periods of rapid conjugate eye movements occurring during sleep were associated with a high incidence of recall of dream content. Sleepers awakened during or shortly after the termination of REMs reported having dreamed, whereas those awakened in the absence of REMs seldom recalled dreaming [2]. The subjective estimation of dream duration was later correlated with the length of the period with REMs [3], and eye movements per se appeared to be related to the content of the dream, suggesting a relationship between the actual imagery of the dream and the direction or number of REMs [4]. Presence of REMs was also associated with the “eventfulness” of the dream story; REMs were supposed to follow the dream content, “scanning” what was happening in the dream. 

Observations of an association between eye movements and dreaming can be found well before the introduction in EEG and EOG recordings. Laad [5], through his own introspection, distinguished between the fixed position of his eyes in deep sleep and their motility during dreaming. Jacobson [6] noted that the eye movements that occur during visual recollection resemble those that occur during dreaming. 

Early studies considered eye movements as the criterion of the presence or absence of dreaming, and a persisting non-spindling, low voltage, moderately fast EEG pattern, defined as “emergent stage 1”, was then associated as invariably concomitant with dreaming “while the amount of eye movement may be large or small depending on the actual imagery of dream” [3]; “active” dreams were also associated with greater ocular activity than “passive” dreams [7]. 

Following observations in animals of the absence of axial tone during period of sleep characterized by low voltage EEG and eye movements [8,9], muscle atonia was introduced as a criterion to define occurrence of “rapid eye movement (REM) sleep” [10], one of the five sleep stages classified in the fundamental Rechtschaffen and Kales manual [11], with the remaining four characterizing the so-called “non-REM sleep” (NREM sleep).

The introduction of the muscle criterion to define REM sleep was an important step in sleep research, since recognition of the “dreaming“ state in initial seminal studies was only based on EEG and eye movements, leading to a potential confounding with awake-drowsy state.

A recent report has reproposed the idea of dream scanning. In patients with REM sleep behavior disorder (RSBD), who appear to enact their dreams due to persistence of muscle tone [12], a directional coherence between limbs, head, and eye movements during REM sleep behavior was found, suggesting that REMs imitate the scanning of the presumed dream scene. 

Andrillon et al. [13] have found that neurons in the medial temporal lobe (MTL), which is correlated with visual awareness, exhibit firing rates during REMs that share many properties with that observed during saccades and vision, suggesting that eye movements during REM sleep rearrange discrete epochs of visual-like processing like those occurring during wakefulness. REMs may constitute privileged time points at which neuronal activity and associated visual-like processing is updated, possibly reflecting a change of the visual imagery in dreams. 

The possible connection between REMs during REM sleep and cognitive content has also been suggested by experimental models in animals. Senzai and Scanziani [14] monitored the head direction (HD) system of the mice thalamus as they explore their environment. They found that in mice, the direction and amplitude of REMs during REM sleep reveal the direction and amplitude of the ongoing changes in head direction in the virtual environments of dreams. According to Senzai and Scanziani, REMs disclose gaze shifts in the virtual world of REM sleep, thereby providing a window in the cognitive process of the sleeping brain.

Although the evidence from Andrillon et al. and Senzai and Scanziani have revitalized the “scanning hypothesis” by showing a relationship between rapid eye movements during sleep and brain events that are also associated with waking vision, the visual-like properties of REMs during REM sleep as indicative of visual imagery of the occurring dream are not supported by consistent findings.

Report of visual dreaming occurrence is also evident during NREM sleep and during the period of tonic REM sleep [15], thus rapid eye movements do not lie necessarily behind the dreaming process. 

Differences between the rapid eye movements during wakefulness and those occurring during the REM periods have been highlighted in several studies (see review of Arnulf [16]) and Moskovitz and Berger [17], Jacobs et al. [18], and Firth and Oswald [19] have not confirmed the eye scan of the dream which was proposed in the early studies. Furthermore, REMs during REM sleep are also present in blind individuals who report no visual dreaming [20], and spontaneous REMs have been found in a hydranencephalic infants where the cortex was presumably absent [21], and REM behaviors have also been reported in a microcephalic infant [22], suggesting that cerebral structures are not involved in REMs. 

## 2. Rapid Eye Movements and Sleep Duration

REM sleep is characterized by rapid eye movements that occur against a stable background of cortical desynchronization and the absence of axial tone. 

During sleep, REM sleep alternates cyclically with NREM sleep in what is considered a functional unit. [23] 

REM density conventionally defines the frequency of oculomotor phasic events during REM sleep, and these REM periods with eye movements are also recognized as “phasic REM”, characterized by bursts of increased EOG amplitudes, whereas the periods without eye movements are classified as “tonic REM”.

In his “Harvey Lecture”, Giuseppe Moruzzi stated: “(…) Summing up, a distinction between underlying tonic changes and phasic outbursts is likely to be useful in any attempt to unveil—trough more refined and more complete electrophysiological and behavioral analysis—the basic nature and the functional significance of sleep” [24].

Usami et al. [25] investigated cortical neuronal responses induced by external inputs during REM sleep. High gamma activity (HGA) suppression during REM sleep was stronger than during wakefulness; however, its suppression during phasic REM sleep was weaker than during the tonic REM sleep period. According to Usami, the change of the cortical excitability in phasic REM sleep was directed toward wakefulness, which may produce incomplete short bursts of consciousness, leading to dreams. 

The different neural states of the phasic and tonic REM periods have been discussed in a recent review by Simor et al. [26]. During phasic periods, external information processing appears to be attenuated [27], probably because brain engages in intrinsically generated cognitive processes. Tonic periods appear as intermediate states between phasic REM sleep and wakefulness with respect to external information processing. 

Increased REMs during REM sleep and REM density have also been reported following intensive learning periods [28,29], suggesting that REM sleep mechanisms might operates off-line memory processing. 

As an alternative to the idea which links REMs during REM sleep to cognitive content, the frequency of oculomotor phasic events occurring during REM sleep appears instead to have its own regulation, which is also separated by the regulation which lies behind the occurrence and duration of the REM periods.

Aserinsky [30] analyzed the sleep of ten young adult subjects who were permitted to sleep to their fullest capacity in a 30-h period. The first four REM periods increased progressively in length but the fifth and sixth were shorter. REM density progressively increased with each successive REM period, with a sharp rise between 7.5 and 10 h of sleep. Additional sleep did not change the REM density that remained at essentially the same level. The extension of sleep beyond the usual quota resulted in an increase in the number of REMs. According to his results, Aserinsky suggested that REM density reflects the satisfaction of a sleep need or the build-up of a pressure to awaken and that since REM density approaches a maximum value with 7.5–10 h of sleep, the leveling-off of REM density as the night proceeds may serve as an index of sleep satiety. 

Benoit et al. [31] studied the temporal organization of periods of sleep with rapid eye movements in ten normal volunteers. The time evolutions of the rapid eye movements and the duration of the REM period resulted in being relatively independent. Across the sleep period, the duration increased more quickly than the number of eye movements.

Consistent with Aserinsky, Feinberg [32] also reported a regular increase in eye movement density across sleep cycles during normal sleep and a striking increase in REM density in late REM periods of extended sleep. Zimmermann et al. [33] studied the sleep of subjects in a free running condition and compared it with the sleep in the entrained condition. REM density increased across successive REM episodes in both conditions, while REM sleep showed a phase advance in its rate of accumulation, with a longer duration of the earlier episodes coinciding with the temperature minimum in the free running condition compared with the entrained condition. Wehr et al. [34] also reported increased REM density during extended sleep in a 10:14 light dark condition. Benson and Zarcone [35] studied rapid eye movement measures in schizophrenics, depressives, and nonpsychiatric controls. Eye movement density increased across REM periods in the schizophrenics and nonpsychiatric controls, whereas the depressives showed a flatter within-night distribution which was attributed to the older age of this group of patients.

Using an automated analyzer for individual eye movements, Takahashi and Azumi [36] measured REMs during REM sleep from 40 nights of polysomnography. REM episode duration and REM density increased progressively in successive REM episodes. The automated parameters describing eye movements magnitude peaked in the second episode followed by relatively low values, thus producing an inverted V pattern. According to Takahashi and Atsumi, the discrepancy between conventional and their automated measures could indicate physiological mechanisms not revealed by conventional measures of REM sleep intensity.

## 3. REM Density after Sleep Deprivation 

Manipulations of the sleep/wake rhythm have been used to assess the significance and the regulation of sleep and of its components. Sleep deprivation either as total night deprivation, SWS deprivation, or REM deprivation have all produced significant reduction in REM density in the recovery night which followed the sleep deprivation.

Antonioli et al. [37] showed that increased pressure for REM sleep produced by selective REM sleep deprivation is manifested by a decreased REM latency and increased REM percent during the recovery night but not by an increased REM density, which instead is reduced during the recovery night. Endo et al. [38] selectively deprived REM sleep in eight healthy participants for three consecutive nights, and a higher amount of REM sleep occurred in the first recovery night, whereas REM density was below the baseline level. Reynolds et al. [39] found a significant decrease in REM density on the first recovery night after sleep deprivation, a finding confirmed by Feinberg et al. [40], who observed that one night’s sleep loss increased the level of slow-wave activity, which is an indicator of sleep pressure, and reduced REM density in recovery sleep, suggesting that the change was due to the greater depth of sleep in the recovery.

In a successive study, Feinberg et al. [41] awakened participants after 100 min of sleep, a duration that includes the first cycle. The analysis of recovery sleep showed eye movement density to be significantly reduced in the second and third REM periods but to a lesser degree than after total sleep deprivation, a condition that, according to Feinberg et al., may be presumed to produce a greater increase in sleep depth. Lucidi et al. [42] studied the sleep pattern of six healthy participants following a gradual sleep restriction, and sleep onset times were progressively postponed, whereas the final awakening time was maintained constant, and they found that sleep curtailment decreases REM density in the recovery nights; the decrease was linearly related to the amount of sleep curtailment. The decrease in REM density also paralleled an increase in SWS, while no corresponding variation was found either in the duration of paradoxical sleep or in the latency of any other sleep stage. De Gennaro et al. [43] studied REM frequency variations in the recovery night after two consecutive nights of selective slow-wave sleep (SWS) deprivation. An independent index of sleep depth, the auditory arousal threshold (AAT), was also studied. In the recovery night, a significant SWS rebound was found, accompanied by an increase in AAT, and REM frequency decreased significantly compared with baseline. According to the authors, the effects cannot be attributed to a variation in prior sleep duration since there was no sleep loss during the selective SWS deprivation nights. Stepwise regression also showed that the decrease in REM frequency was not correlated with the increase in AAT but was correlated with SWS rebound. Marzano et al. [44] also studied the effects of sleep deprivation on REM density. They found a clear reduction in REM density in the recovery night, although REM densities did not change across cycles. Oculomotor changes positively correlated with a decreased power in a specific range of fast sigma activity (14.75–15.25 Hz) in NREM but not with SWA. REM density changes were also related to EEG power in the 12.75–13.00 Hz range in REM sleep. In contrast with previous studies, the depression of REM density appeared to not be related to homeostatic mechanisms since REM density changes were associated with EEG power changes in spindle frequency activity but not with SWA. Darchia et al. [45] reported that in baseline sleep, eye movement density is not correlated with rates of non-rapid eye movement (NREM) delta production (power/min), also, daytime napping does not produce changes in eye movement density and delta power. According to Darchia et al., very strong changes in sleep depth are required to overcome the individual stability of NREM delta and eye movement density. 

The effects of sleep deprivation on REM density in the recovery night were also confirmed in habitual short sleepers (sleep episode < 6 h) and long sleepers (sleep episode > 9 h). Sleep deprivation decreased sleep latency and rapid eye movement (REM) density more in long sleepers than in short sleepers [46]. Consistently, with the hypothesis of an inverse relationship between REM density and the depth of sleep, the enhancement of EEG slow-wave activity (SWA; spectral power density in the 0.75–4.5 Hz range) in non-REM sleep after sleep loss was larger in long sleepers (47%) than in short sleepers (19%).

## 4. Circadian Modulation of REM Density

REM sleep duration has a circadian distribution [47,48], peaks in REM sleep propensity have been shown to occur in proximity of the minimum of body temperature curve. From early studies, it appeared that REM density would also follow a circadian regulation. 

Foret et al. [49] showed that in individuals submitted to an unusual schedule, with sleep starting between 2 and 5 a.m., the REM periods have few eye movements and are of short duration, suggesting an effect of clock time. Kobayashi et al. [50] reported a circadian distribution of REM density in scheduled sleep opportunities following wakefulness lasting either 16, 26, 30, or 34 h. REM density was low after 16 h and highest after 26 h in the morning and then decreased in the evening after 30 and 34 h. Witzenhausen et al. [51] examined REM density distribution in scheduled day time naps and in night sleep. A double nap procedure was adopted: nap A, where sleep was restricted to 30 min and then the subject was awakened, followed by nap B, of variable length with recording finished 15 min after the last REM epoch was recorded. REM density showed a steady decrease from morning to late evening in the naps scheduled at 2 h intervals between 0800 and 2400. During night sleep, REM density increased from the first to the second and last thirds of the night. REM density paralleled the circadian distribution of tonic REM sleep parameters. Although both Kobayashi et al.’s and Witzenhausen et al.’s studies suggest a circadian variation of REM density, the research models used in the two studies do not allow us to exclude the rule of prior wakefulness that might have influenced REM density variation.

Khalsa et al. [52] analyzed REM density in a forced desynchrony protocol [53], which can distinguish the sleep and circadian dependent modulation of sleep regulation. The forced desynchrony protocol consisted of an activity/rest schedule with 18 h 40 min of wakefulness in dim light and 9 h 20 min sleep in darkness. Twenty consecutive sleep opportunities were systematically scheduled over the entire circadian cycle. REM density showed a robust increase over the course of the sleep episode. Data analyzed from different thirds of the circadian cycle exhibited a similar pattern, suggesting a non-significant circadian modulation, although data from the last third of the sleep episode showed that the maximum REM density coincided with the wake maintenance zone, a result not in agreement with Kobayashi et al.’s results that, however, were obtained using a different study model. Results from Khalsa et al.’s study are consistent with the idea of sleep-dependent modulation of REM density, the frequency of REMs is greatest in the latter part of the sleep episode, when sleep pressure is low. 

## 5. REM Density and Spontaneous Awakening

One of the hypotheses relative to the function of REM sleep is that it may provide a “gate” for wakefulness [54]. Most spontaneous awakenings occur out of REM sleep [55,56], and Broughton found that the visual evoked potentials of subjects after a forced arousal from REM sleep were like those of wakefulness, whereas subjects aroused from stage 4 presented slower evoked potentials [57]. Subjects instructed to wake at a specific time without the aid of an external alarm tended mainly to awake from REM sleep [58].

Barbato et al. [59] analyzed the sleep pattern of individuals exposed for 4 weeks to a winter type photoperiod of 10 h light and 14 h darkness. The prolonged period of darkness allowed sleep extension with a contemporary reduction in the wake period resulting in a diminished sleep pressure. They found that the propensity to wake from sleep was higher in the REM sleep period with a high density of REM than from the NREM sleep period, possibly reflecting an increased level of the brain arousal process associated with REM sleep. In a successive paper [60], they confirmed that in extended nights the REM densities of the REM periods that terminated in periods of wakefulness were higher than those of uninterrupted REM periods, although the level of REM density was not able to predict the duration of the sleep interruption that followed. The proportion of episodes of wakefulness following REM sleep that were long-lasting progressively increased over the course of the extended night period, suggesting a combined effect of reduced sleep pressure and increased arousal. Campbell [54] has suggested that spontaneous transitions from REM sleep to wakefulness tend to occur when the neural activity that induces REM sleep is increasing exponentially in association with phasic activation [61,62]. Lowering of SWS propensity due to chronic sleep satiety in the extended sleep paradigm might also have facilitated the transition to wakefulness.

Ficca et al. [63] compared REM densities and occurrence of awakening in three different age groups: young (age range 20–25 years), old (age range 62–74 years), and very old subjects (age range 77–98 years). Whereas in the young, REM density was higher in the REM periods followed by spontaneous awakening, the difference was reduced in the old group and absent in the very old subjects. The evidence reported by the authors that old individuals spontaneously wake up despite the absence of an increase in REM activity could imply that in older subjects awakening is not preceded by an increase in the arousal level. Vegni et al. [64] reported significantly reduced eye movement density in a small sample of elderly participants, data which were confirmed by Darchia et al. [65], who reported that the incidence of eye movements during REM sleep is substantially reduced in the elderly. The authors hypothesized that this is due to degenerative (aging) changes. Fein et al. [66] reported a lower incidence of visual imagery in dream reports following NREM and REM forced awakenings in elderly subjects, a finding that suggests that the reduced visual imagery cannot be attributed to a reduced frequency of eye movements.

Interestingly, alteration of the frequency of REM during REM sleep was also reported in relation to cognitive performance. Feinberg et al. [67] first showed that low REM activity in healthy older adults was associated with lower performance on psychometric tests. Spiegel et al. [68] reported that low REM density predicted the level of cognitive decline in older subjects. A reduction in REM density and of the other phasic components of REM sleep has been reported in vegetative state patients, suggesting possible damage to the pedunculopontine tegmentum cholinergic mechanisms [69]. Early studies in the 1960s documented a reduction in bursts of REMs in mentally retarded children, leading to the idea of REM density as an index of intellectual level [67]; reduction in REM density has also been reported in Alzheimer’s dementia and suggested as a sensitive biological marker for the differential diagnosis between old-age depression and Alzheimer’s [70].

## 6. REM Density and Depression

Sleep alterations are a typical feature of depressive symptomatology, and patients frequently report disturbed, fragmented, non-restorative sleep. Polysomnographic recordings have highlighted differences in the sleep patterns of depressive patients compared to those of healthy subjects. Sleep in depressive patients is characterized by reduced sleep efficiency, reduction in slow wave sleep, reduced REM sleep latency, increased REM density, increased frequency of awakenings during the sleep period, and early morning awakening.

Studies in the 1970s suggested short REM latency as a biological marker of major depressive illness, capable of differentiating primary from secondary depression [71]. Successive studies have, however, not confirmed the specificity of short REM latency in depression since it can be found in several other major psychiatric conditions [72]. As suggested by Buysse and Kupfer [73], clinical validity of polysomnographic measures in depressive patients may be affected by the competing needs of clinical vs. research studies and by focusing too narrowly on the diagnostic use of sleep measures.

The other REM measure, REM density, was then proposed as a sensitive and reliable indicator of depressive clinical status, as a risk factor for the emergence of depression, and as a predictor of the patient response to treatments [74].

Kupfer and Heninger [75] reported variation in REMs during REM sleep correlated to mood changes in a manic-depressive patient. On nights preceding depressive phases, the patient showed more early morning awakening and less REM activity compared to nights preceding hypomanic phases.

REM density has been found significantly higher in the first REM period in bipolar disorder patients compared to healthy controls [76,77] and is also frequently associated with a contemporary reduction in slow wave sleep.

Borbely and Wirz-Justice [78] have hypothesized that alteration of sleep in depression can be explained according to the two process model of sleep regulation [79] as the consequence of a dampened and altered time course of slow wave sleep, and thus REM density would be higher in the first cycle because of the reduced pressure of slow wave sleep.

According to Arfken et al.’s [80] meta-analysis, REM density is consistently higher in patients with major depression compared with controls, showing an overall mean of 1.94 for the patient group vs. 1.63 for the controls.

Alteration of REM density was also described as a potential indicator of vulnerability to major affective disorders. A positive relation between Zung depression scores and eye movement density during REM sleep was reported in a sample of 19 noncomplaining young adult males [81]. In a study specifically designed to assess vulnerability factors for affective disorders, Lauer et al. [82] in a sample of healthy subjects with a high genetic load for psychiatric disorders, reported a reduced amount of slow wave sleep and increased REM density in the first sleep cycle. During the follow-up period, 20 of the previously identified healthy high-risk probands developed an affective disorder, and their premorbid sleep was characterized by an increased REM density both in the total night and in the first REM period compared with the control group without history of a psychiatric disorders [83]. According to their data, the authors suggested that an increased REM density can be considered as a possible endophenotype for affective disorders.

Thase et al. [84] used an index derived from reduced sleep efficiency, reduced REM latency, and increased REM density to predict response to treatment. Patients with a diagnosis of recurrent major depressive disorder showing this abnormal sleep profile were less responsive to psychotherapy. According to the authors, a “neurobiological boundary” may limit response to psychotherapy in depression, and the abnormal sleep profile may reflect a marked disturbance of CNS arousal that can benefit from pharmacotherapy. Buysse et al. [85] reported that elevated phasic REM sleep predicted patients who did not respond (non-remitters) to interpersonal psychotherapy (IPT), and also, the accumulation of REM activity during sleep occurred at a significantly higher rate in non-remitters than in remitters. In a successive study [86], IPT non-remitters at baseline before the treatment had increased phasic REM sleep compared with remitters but had no significant differences in EEG power spectra. Patients who recovered with a combined treatment of IPT and fluoxetine had an increase in phasic REM. According to Buysse et al., the number of REMs was a more robust correlate of remission and recovery than quantitative EEG during NREM or REM sleep. REM phasic activity may thus provide a more direct measure of brainstem function during REM sleep than quantitative sleep EEG measures.

Consistent with these data, Lechinger et al. [87] found that higher baseline REM density predicted better response to antidepressant pharmacotherapy. According to Lechinger et al., patients with a disturbed phasic REM sleep may have an overactive emotional arousal system and might benefit from antidepressant medication, which suppresses REM sleep and reduces hyperactivity in various networks related to emotional memory, valence, and arousal. Goder et al. [88] reported that patients with major depressive episode who responded to IPT showed a trend towards a decrease in REM density during the first REM period and a significant decrease in delta power in comparison to non-remitters. Patients with higher REM density during the first REM period have also shown better response to electroconvulsive therapy than patients with lower REM density [89].

Increased REM density has also been reported in other clinical conditions where a disorder of the central arousal mechanisms is hypothesized. Considering REM density a measure of sleep need, Vankova et al. [90] studied patients with idiopathic hypersomnia and narcolepsy, both characterized by excessive daytime sleepiness, in an effort to use REM density as a possible tool for differential diagnosis. REM density was higher in both groups of hypersomnia patients in comparison with control; however, no significant difference between the two groups of patients was found.

Narcolepsy and REM sleep behavior disorder (RBD) present characteristic dissociated sleep/wake states that are attributed to decreased hypocretinergic and/or dopaminergic abnormalities input to brainstem structures. In a study [91] assessing REM sleep characteristics in these two conditions, REM density was found higher in narcoleptic patients, and lower in RBD, although both groups of patients showed a high percentage of REM without atonia, suggesting the existence of different pathway in the regulation of EOG and EMG phasic components. 

According to Dauvilliers et al., disinhibition of both phasic activities in narcolepsy is due to a common neurobiological defect, i.e., hypocretin, whereas in RBD may be related to selective damage within brainstem that avoids the structure controlling eye movements [91]. 

As for depressed patients in patients with post-traumatic stress disorder, a dysfunction of REM sleep has been also hypothesized. A meta analytic review of 20 studies showed that PTSD patients had greater REM density and more stage 1 and less slow wave sleep compared to individuals without PTSD [92]. Increased REM density has been reported in chronic PTSD patients with a duration of decades [93]. A recent study compared PTSD patients with patients with major depressive disorder. REM density was significantly increased in the PTSD group compared to the MDD group. The severity of trauma related nightmare complaints significantly correlated with the percentage of REM interruption but not REM density [94].

## 7. Discussion

The functional significance of REMs during the REM sleep period is unknown and not well defined are the factors regulating their occurrence throughout sleep.

Although fascinating, the hypothesis that REMs during sleep are directly related to dreaming processes and may provide a window into the mental activity of the dreamer has not been supported by consistent findings. Recent studies also suggest that dreaming and REM sleep are dissociable states and that dreams can be related to forebrain mechanisms [95,96], in contrast with the traditional view that attributes the generation of dreams to brainstem mechanisms [97].

Data on the relationship between frequency of REMs during the REM period and level of sleep pressure, as originally proposed by Feinberg et al. [40], seem instead to favor the idea that links REMs during REM sleep to brain arousing mechanisms being the overt expression of increased activity during REM sleep [98]. In the present review, some evidence from different experimental conditions and from clinical data support this view.

Sleep pressure, as measured by slow wave sleep and/or slow wave activity, diminishes across the night due to the progressive release of cumulated sleep need, whereas sleep pressure is increased by prior total and partial sleep deprivation [99]. REM density instead increases across the night concurrently with the progressive reduction in sleep depth, and it is higher at the circadian time when arousal appears to be higher, and it is decreased in those conditions, such as after sleep deprivation, which produce increased sleep pressure.

Data on depression, in which a condition of hyperarousal is hypothesized [100,101], are consistent with this idea. REM density is higher in major affective disorder and in subjects with a familial history of affective disorder, and the presence of a higher REM density is also predictive of a positive response to somatic treatment, suggesting an underlying disturbance of the central nervous system.

Neuroimaging studies in depressed patients have shown higher cortical and limbic metabolic rates, including in the amygdala and the posterior cingulate, during sleep [102]. Amygdala overactivity in depression has also been described during wakefulness [103] and suggested as responsible of increased emotional arousal [104]. In depressed patients, Nofzinger et al. [105] have shown increased activation in paralimbic and executive cortex from waking to REM, which is suggestive of affective and cognitive dysregulation. According to Van der Helm et al. [106], REM sleep is associated with an overnight dissipation of amygdala activity in response to previous emotional experiences, altering functional connectivity and reducing next-day subjective emotionality.

Recent findings have identified “restless REM”, a condition of fragmented REM sleep with a high frequency of eye movements, as an important marker of insomnia [107]; furthermore, in this patient group, higher REM density, together with REM arousal, was strongly associated with the slow dissolution of emotional distress.

Interestingly, an early study [108] showed that a psychologically stressful film seen just before bed produced a greater proportion of REM period terminated by spontaneous awakenings and an increased REM density, suggesting a disturbing effect via anxiety.

Schroeder et al. [109] used REM density as a marker of sleep pressure in Parkinson disease. REM density was reduced in patients with idiopathic Parkinson disease (IPD) compared with healthy controls, possibly reflecting direct involvement of the brainstem REM generation sites by the disease process. The REM density reduction correlated with subjective score sleep impairment, suggesting that as an indicator of persistent high sleep pressure, reduced REM density in IPD can be utilized as a marker of excessive daytime sleepiness.

Severe reduction in REM density has also been reported in patients with spinocerebellar ataxia type 2, [110] a disorder of progressive atrophy from the pons, nigrostriatal projection, and locus coeruleus to the thalamus. Reduction in REM density correlated with severity and disease progression.

Since the early work of Pompeiano and Morrison [111], REM phasic activity has been related to control mechanisms in the brainstem. REM sleep is generated by cholinergic projections from the pedunculopontine tegmental nuclei (PPT) that activate neurons in the nucleus pontis oralis, and PPT is also under control from excitatory (glutamatergic) projections from the central nucleus of the amygdala [112]. Increased cholinergic activity and reduced monoamine (serotonin and noradrenaline) activity from the forebrain and midbrain are considered responsible of the cortical activation during REM [113]. Machowsky [114] has suggested that REM sleep and wake could constitute two steps of an arousal process controlled by cholinergic mechanisms. Interestingly, experimental studies in humans have shown that physostigmine, a reversible anticholinesterase, if infused during NREM sleep, induces REM sleep, and if infused during REM sleep, it induces wakefulness [115] and that scopolamine, a potent anticholinergic, reduces REM phasic activity in humans [116].

REMs during REM sleep are hypothesized to be produced by mechanisms like those suggested for PGO waves [117]. In animal studies, REMs during REM sleep are closely related to ponto–geniculo–occipital (PGO) waves, phasic activities that have been recorded from the pons, lateral geniculate, and the occipital cortex [118,119]. Although PGO waves have not been recorded in humans, using a positron emission topography and iterative cerebral blood flow by H(2)(15)O infusions, Peigneux et al. [120] found a significant interaction effect in the right geniculate body and in the primary occipital cortex during REM sleep and in relation with the density of REM.

In a recent paper, Saleh et al. [121] have emphasized the necessity to reconsider REM density in sleep research and to be utilized as a biomarker for both diagnostic precision and prognosis. The proposed relationship of REM density with arousal reviewed in the present paper may provide an easily measurable and reliable parameter to investigate different clinical conditions where a disorder of the arousal system is hypothesized as well as to assess and define treatment for these disorders.

Although previous studies have emphasized the role of acetylcholine and glutamate in arousal, recent evidence has also highlighted the role of dopamine and hypocretin in the regulation of sleep wake mechanisms, providing a further challenge for understanding the functional significance and regulation of REM sleep and of its phasic component.

As occurred in early reports, REM density can be an index of the intensity of REM sleep and used to better characterize and define an important yet “unknown” part of the sleep phase. The opposite behavior of slow wave sleep and REM density across the night further suggest that the two components of the sleep cycle, NREM and REM sleep, can be considered as separate antagonistic process, respectively governing the first and the second halves of night sleep. The relationship between increased REM density and awakening that has been described in extended sleep suggest that REM density can be utilized to explore those mechanisms which end sleep, and a frequency range could provide a physiological marker which indicate during sleep the “time to wake”.

The seminal idea of Aserinsky, that REM density reflects the amount of prior accumulated sleep and that it may serve as an index of “sleep satiety”, needs to be further explored and confirmed. REM density may constitute a sensitive measure of sleep homeostasis in addition to, or even as an alternative to, the consolidated analysis of slow wave activity.

## Data Availability

Not applicable.

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
