# Peer review of "Is REM Density a Measure of Arousal during Sleep?"

_brainsci, 2023, doi:10.3390/brainsci13030378_

Round 1

Reviewer 1 Report

This is an interesting and clearly written review exploring the functional significance of REM density and the potential role of this as a biomarker to predict the development of affective disorders or the response to treatment.

Comments:

  • The review could benefit from the inclusion of figures.  For example, a hypnogram showing REM distribution, EEG traces showing low and high REM density, or a diagrammatic representation of the role of REM density in normal sleep and in affective disorders
  • It would be preferable to replace “subject” with “participant” throughout
  • The majority of references are quite old (only six references out of 83 were published since 2011) and whilst the older references are relevant to the topic, more current references should be included particularly in relation to clinical conditions
  • The first paragraph of section 2 would read better at the end of section 1
  • Lines 89 – 98: can it be clarified that the REM density remained the same in the shorter fifth and sixth REM periods?  Which REM period occurred between 7.5 and 10 hours?
  • Line 92 and 97 should match with 7.5 or 7.7 hours?
  • Lines 107 – 110: how should this advance in free-running conditions be interpreted?
  • Lines 116 – 122: the text relating to reference [25] states that the authors discuss the difference “between conventional and new measures” but this paper was written in 1997 and so the text should be updated
  • Lines 178 – 180: this is repetitive – also explain where the Tmin occurs in relation to wake
  • Lines 181 – 183: this seems out of context and should be linked to the previous sentence
  • Lines 184 – 186: what was the circadian time of these episodes?  How long did they sleep for?
  • Lines 186 – 196: how long were these nap opportunities?
  • Line 198: how long were the sleep opportunities and what was the T cycle?
  • Line 204: maximal REM density occurred in the wake maintenance zone – this should be contrasted to the work of Witzenhausen showing a decrease in REM during evening naps
  • Lines 232 – 233: what were the age ranges of these groups?
  • Lines 265 – 285: these read as bullet points and need combining into a cohesive stream, and interpreted collectively
  • The Discussion reads as a series of bullet points and should be reformulated, and a conclusion included

The manuscript has some minor typos throughout e.g., line 60 should be “of the dream”.

Author Response

This is an interesting and clearly written review exploring the functional significance of REM density and the potential role of this as a biomarker to predict the development of affective disorders or the response to treatment.

Comments:

  • The review could benefit from the inclusion of figures.  For example, a hypnogram showing REM distribution, EEG traces showing low and high REM density, or a diagrammatic representation of the role of REM density in normal sleep and in affective disorders.

       A figure showing REM density has been added.

  • It would be preferable to replace “subject” with “participant” throughout.

It has been replaced, where appropriate

  • The majority of references are quite old (only six references out of 83 were published since 2011) and whilst the older references are relevant to the topic, more current references should be included particularly in relation to clinical conditions.

Several new references, also related to clinical conditions have been added.

  • The first paragraph of section 2 would read better at the end of section 1

      The paragraph has been moved accordingly.

  • Lines 89 – 98: can it be clarified that the REM density remained the same in the shorter fifth and sixth REM periods?  Which REM period occurred between 7.5 and 10 hours?

This information was not available in the cited paper.

  • Line 92 and 97 should match with 7.5 or 7.7 hours.

       Yes, it was a typo (7,5), thanks.

  • Lines 107 – 110: how should this advance in free-running conditions be interpreted?

“Coinciding with the temperature minimum in the free running condition” added in the revised version, lines 133

  • Lines 116 – 122: the text relating to reference [25] states that the authors discuss the difference “between conventional and new measures” but this paper was written in 1997 and so the text should be updated.

The statement has been corrected accordingly, lines 140-147

  • Lines 178 – 180: this is repetitive – also explain where the Tmin occurs in relation to wake
  • Lines 181 – 183: this seems out of context and should be linked to the previous sentence

The statement has been changed and the two-sentence linked,

  • Lines 184 – 186: what was the circadian time of these episodes?  How long did they sleep for?

 This information was not available in the cited paper.

  • Lines 186 – 196: how long were these nap opportunities?
  • Line 198: how long were the sleep opportunities and what was the T cycle?

The information has been added, the T cycle was not available since the authors used melatonin to characterize the circadian rhythm.

  • Line 204: maximal REM density occurred in the wake maintenance zone – this should be contrasted to the work of Witzenhausen showing a decrease in REM during evening naps.

Yes, a statement discussing the point has been added.

  • Lines 232 – 233: what were the age ranges of these groups?

The information has been added. 

  • Lines 265 – 285: these read as bullet points and need combining into a cohesive stream, and interpreted collectively.

The section has been expanded, and further discussed.

  • The Discussion reads as a series of bullet points and should be reformulated, and a conclusion included.

The discussion has been largely edited, and several points have been added.

The manuscript has some minor typos throughout e.g., line 60 should be “of the dream”.

This was not a typo, the cited authors used “of the REM sleep”, the study was on animals.

Reviewer 2 Report

Thanks to the authors of this article...

My main challenge with this article is what is the general conclusion? This article has no conclusion, and I don't understand what message the authors are trying to convey? Although they have hints, they are very general, and do not mention anything new. Until the discussion, all texts refer to previous works, which are not new points. I was waiting for an important topic to be mentioned in the discussion, but it was almost a repetition of the previous content. And at the end, there was no conclusion.

The topic of the article is very interesting to me. Thanks to the authors for choosing this title. Our knowledge of sleep is very limited, and the REM phase is one of the most fundamental stages of sleep that is involved in many of our cognitive and psychological activities. In this article, many discussions are left out, and others are only lightly mentioned.

For example, on page 2, line 50-54, an important topic was mentioned, but you explained it very narrowly in four lines. Or on page 2, line 64-75, you mentioned an interesting issue, please discuss more. Or page 7, line 351. Or page 7, line 340. and...

Throughout the article, authors did not sufficiently mention the neurobiology of sleep. The authors also did not mention the important role of neurotransmitters (except in very few cases). Furthermore, changes in REM density and its role in cognitive activities have not been explained. How is REM density modulated by the brain? And what is the cause of its changes in mental disorders? What about REM density changes in neurodegenerative disorders? Or other disorder? Are there functional changes for any brain region following changes in REM density? Or for neurotransmitter systems?

The article, contrary to its fascinating title, is very limited. Please open more discussions...

Author Response

Thanks to the authors of this article...

My main challenge with this article is what is the general conclusion? This article has no conclusion, and I don't understand what message the authors are trying to convey? Although they have hints, they are very general, and do not mention anything new. Until the discussion, all texts refer to previous works, which are not new points. I was waiting for an important topic to be mentioned in the discussion, but it was almost a repetition of the previous content. And at the end, there was no conclusion.

The discussion has been largely edited, and several points have been added, lines 454-472 discuss possible topic for use of REM density measure.

The topic of the article is very interesting to me. Thanks to the authors for choosing this title. Our knowledge of sleep is very limited, and the REM phase is one of the most fundamental stages of sleep that is involved in many of our cognitive and psychological activities. In this article, many discussions are left out, and others are only lightly mentioned.

For example, on page 2, line 50-54, an important topic was mentioned, but you explained it very narrowly in four lines.

Further information on the cited work were added, lines 52-58,

Or on page 2, line 64-75, you mentioned an interesting issue, please discuss more.

Lines 72-74, have been added, Among the others, one of the references is a review article largely discussing the topic, the issue is then analyzed in the revised discussion, lines 390-395,

Or page 7, line 340.

 Or page 7, line 351.

This part has been expanded in the revised discussion, lines 435-451

Throughout the article, authors did not sufficiently mention the neurobiology of sleep. The authors also did not mention the important role of neurotransmitters (except in very few cases).

In the revised paper, several statements have been added discussing these points: 349-385, 367-376, lines 396-400, 401-406, 435-451

Furthermore, changes in REM density and its role in cognitive activities have not been explained.

In the revised paper REM density and cognitive activities have been added: lines:  96-101, 102-109, 277-287

How is REM density modulated by the brain? And what is the cause of its changes in mental disorders? What about REM density changes in neurodegenerative disorders? Or other disorder?

In the revised paper changes in REM density in neurological disorders and other disorders have been added, lines 360-385, 428-438,

Are there functional changes for any brain region following changes in REM density? Or for neurotransmitter systems?

Neurotrasmitter systems have been further discussed, lines 439-459

The article, contrary to its fascinating title, is very limited. Please open more discussions...

The discussion has been largely edited, and several points have been added.

Reviewer 3 Report

In the manuscript entitled  "Is REM density, a measure of arousal during sleep? author tried to review and discuss about how REM density could be used/consider measuring arousal during sleep. Author has just described what the other peoples have found but unfortunately did not talk/discuss about what does their actual meaning is? None of the section including discussion fully justify the title. Discussion section is poorly written and does not justify the title.  

Author Response

In the manuscript entitled  "Is REM density, a measure of arousal during sleep? author tried to review and discuss about how REM density could be used/consider measuring arousal during sleep. Author has just described what the other peoples have found but unfortunately did not talk/discuss about what does their actual meaning is? None of the section including discussion fully justify the title. Discussion section is poorly written and does not justify the title.  

The discussion has been largely edited, and several points have been added, lines 454-472 discuss possible topic for use of REM density measure.

Round 2

Reviewer 2 Report

The author has done a good job of responding to most points of clarification. I have no concern and the revised version is acceptable.

Author Response

THANKS